# Proteasome-Associated Proteins, PA200 and ECPAS, Are Essential for Murine Spermatogenesis

**DOI:** 10.3390/biom13040586

**Published:** 2023-03-24

**Authors:** Ban Sato, Jiwoo Kim, Kazunori Morohoshi, Woojin Kang, Kenji Miyado, Fuminori Tsuruta, Natsuko Kawano, Tomoki Chiba

**Affiliations:** 1Master’s and Doctoral Program in Biology, Faculty of Life and Environmental Sciences, University of Tsukuba, 1-1-1 Tennodai, Tsukuba 305-8577, Japan; 2Laboratory of Regulatory Biology, Department of Life Sciences, School of Agriculture, Meiji University, 1-1-1 Higashimita, Kawasaki 214-8571, Japan; 3College of Biological Sciences, School of Life and Environmental Sciences, University of Tsukuba, 1-1-1 Tennodai, Tsukuba 305-8577, Japan; 4Department of Reproductive Biology, National Research Institute for Child Health and Development, 2-10-1 Okura, Setagaya 157-8535, Japan

**Keywords:** proteasome, PA200, ECPAS, LPIN1, spermatogenesis, sperm anomalies

## Abstract

Proteasomes are highly sophisticated protease complexes that degrade non-lysosomal proteins, and their proper regulation ensures various biological functions such as spermatogenesis. The proteasome-associated proteins, PA200 and ECPAS, are predicted to function during spermatogenesis; however, male mice lacking each of these genes sustain fertility, raising the possibility that these proteins complement each other. To address this issue, we explored these possible roles during spermatogenesis by producing mice lacking these genes (double-knockout mice; dKO mice). Expression patterns and quantities were similar throughout spermatogenesis in the testes. In epididymal sperm, PA200 and ECPAS were expressed but were differentially localized to the midpiece and acrosome, respectively. Proteasome activity was considerably reduced in both the testes and epididymides of dKO male mice, resulting in infertility. Mass spectrometric analysis revealed LPIN1 as a target protein for PA200 and ECPAS, which was confirmed via immunoblotting and immunostaining. Furthermore, ultrastructural and microscopic analyses demonstrated that the dKO sperm displayed disorganization of the mitochondrial sheath. Our results indicate that PA200 and ECPAS work cooperatively during spermatogenesis and are essential for male fertility.

## 1. Introduction

Protein degradation is necessary to maintain homeostasis [1]. The proteasome is a 2.5-MDa multi-subunit protease complex that degrades non-lysosomal proteins, and its proper regulation guarantees the success of various biological events, such as the cell cycle, immune response, and gametogenesis [2,3,4]. The 26S proteasome consists of two subcomplexes, a 20S catalytic core particle (CP) and 19S regulatory particles (RPs), that bind at either or both ends of the 20S CP [5]. Notably, 20S CP is a barrel-shaped complex consisting of two inner β-rings and two outer α-rings stacked with seven different subunits. Although 20S CP is constitutively expressed in a wide variety of tissues, three tissue-specific forms have been recently discovered: immunoproteasomes in T cells, thymoproteasomes in the thymus, and spermatoproteasomes in the testes [6,7]. In mammalian testes, proteasomes specialize in spermatoproteasomes that contain testis-specific 20S subunit α4s/PSMA8 and/or catalytic β subunits of immunoproteasomes [8,9]. Since the catalytic activity and type of protein substrate of spermatoproteasomes differ from those of other proteosomes, the combination of 20S and its regulators, including RPs, has been the focus of research.

Constitutive 20S CP is regulated by various RPs, such as PA700 (19S RP), PA28 (11S), and PA200/Blm10, to open the α-ring gate. PA700 captures ubiquitinated and unfolding proteins by using ATPase [8,10]. In contrast, the PA200/Blm10 (a yeast homolog of murine Psme4) regulator degrades unstructured and acetylated proteins in the absence of ubiquitination and an ATP-utilizing system [10,11,12]. In addition to RPs and CP composition, various accessory proteins, such as PI31 and ECPAS (Ecm29), contribute to the regulation of proteasome activity [13,14,15]. ECPAS inhibits proteasome activity by binding to the CP–RP complex and acts as a checkpoint for CP maturation. As mentioned above, multiple components tightly control proteasome function.

Spermatogenesis is regulated by various proteasome components [16]. *α4s*-deficient mice grew up healthy but exhibited infertility due to meiotic failure caused by dysregulated proteostasis [8,17]. In RP, PA200-capped proteasomes promote the acetylation-dependent degradation of core histones during somatic DNA repair and spermiogenesis [12,18]. Previously, we reported that *Psme4*-deficient male mice are rendered infertile by double knockout with *Psme3* [19]. Male mice lacking PITHD1, a protein associated with immunoproteasomes, exhibit infertility due to morphological abnormalities in the sperm [20]. Therefore, spermatogenesis requires a tightly regulated proteolytic system. However, the entire mechanism remains unelucidated.

Among the regulators of ECPAS and PA200, a unique homology composed of HEAT-like repeat motifs is found, such as an α-helical solenoid structure [21]. In yeasts, double mutants of these molecules are highly sensitive to proteotoxic stress [10]. A single mutant of these molecules exhibits normal fertility; however, whether their functions are mutually compensated remains unproven in mice. We hypothesize that PA200 and ECPAS act in concert and are functionally complementary. To explore this issue, we generated double knockout (dKO) mice and explored the potential roles of these proteins in male murine fertility.

## 2. Materials and Methods

### 2.1. Animal Experiments

Mice were housed under specific pathogen-free conditions. Food and water were provided ad libitum. All animal experiments were approved by the Animal Care and Use Committee of the Graduate School of Life and Environmental Sciences at the University of Tsukuba (19-292). Ethical issues were addressed in accordance with the approved guidelines of the University of Tsukuba.

### 2.2. Antibodies (Abs)

Anti-PA200 and anti-ECPAS polyclonal Abs developed in our lab are described in previous studies [22]. Anti-α-tubulin monoclonal (Sigma-Aldrich, St. Louis, MO, USA), anti-TOM20 (Santa Cruz Biotechnology, Dallas, TX, USA), and anti-LPIN1 monoclonal (Genetex Inc., Irvine, CA, USA) Abs were used in this study.

### 2.3. Production of dKO Mice

*Psme4* knockout mice (B6.CBA-Psme4^<tm1Tchi>^) and *Ecpas* knockout mice (B6.CBA-Ecm29^<tm1Tchi>^) have been previously described (Appendix A) [12,19,22]. These mouse lines (ID: RBRC09401 and BBRC09402) were deposited at RIKEN BRC, Japan. *Psme4* and *Ecpas* dKO mice were obtained by intercrossing *Psme4*^+/−^*Ecpas*^−/−^ male mice and *Psme4*^−/−^*Ecpas*^−/−^ female mice as breeding pairs.

For murine genotyping, genomic DNA (extracted from the tail tips) was subjected to polymerase chain reaction (PCR) analysis. Wild-type and mutant alleles for each gene were detected using multiplex PCR with three primers (Appendix A).

### 2.4. Proteasome Activity

The chymotrypsin-like activity of proteasomes in the testes and epididymides was determined using a fluorescence substrate, Suc-LLVY-MCA (Peptide Institute, Osaka, Japan). Testicular and epididymal tissue extracts were incubated with 50 μM fluorescence substrate for 30 min at 37 °C. Fluorescence was measured via excitation at 380 nm and emission at 460 nm using a spectrofluorometer (PerkinElmer Inc., Waltham, MA, USA). The results were estimated as fluorescence intensity per protein amount and reaction time.

### 2.5. Histochemistry and Immunohistochemistry

Sperm collected from the epididymides were washed with phosphate-buffered saline (PBS), plated on MAS-coated glass slides, and dried. The sperm were incubated with the MitoTracker^TM^ Green (Thermo Fisher Scientific, Waltham, MA, USA) and peanut agglutinin (PNA) lectin conjugated with Alexa 594 (Thermo Fisher Scientific, Waltham, MA, USA) overnight at 4 °C following PBS washing. Sperm were then mounted on glass slides and observed under a fluorescence microscope (Biorevo, BZ-9000, Keyence, Japan). Cauda epididymides and testes were fixed in a solution containing 4% paraformaldehyde in PBS and embedded in paraffin. Paraffin-embedded sections were subjected to periodic acid–Schiff (PAS) and hematoxylin and eosin (HE) staining. The frozen sections of the testes were used for Oil Red O staining.

### 2.6. Mass Spectrometry

The epididymides were extracted in 50 μL of lysis buffer (8 M urea and 0.2 M ammonium bicarbonate) and sonicated. Each lysate was digested with trypsin and labeled with iTRAQ 4-plex reagent, according to the manufacturer’s instructions. An Orbitrap Q Exactive Plus spectrometer (Thermo Fisher Scientific, Waltham, MA, USA) and an EASY-nLC 1200 chromatograph (Thermo Fisher Scientific, Waltham, MA, USA) were used to generate shotgun proteomics data. Liquid chromatography with tandem mass spectrometry (LC-MS/MS) analysis and database searches were conducted using APRO Science (Tokyo, Japan).

### 2.7. Transmission Electron Microscopy (TEM)

For ultrastructural analysis, mice were perfused with a solution containing 2% glutaraldehyde and 2% paraformaldehyde, and the testes were then dissected, followed by immersion fixation in 2.5% glutaraldehyde in 0.1 M phosphate buffer. The sperm collected from the cauda of epididymides were sunk in 2.5% glutaraldehyde. These samples were then processed according to standard procedures, and ultrathin sections were subjected to TEM analysis using a JEM-1400 electron microscope (JEOL Inc., Tokyo, Japan) at the electron microscopy facility located at the University of Tsukuba. Multiple sections from 1 to 2 mice per genotype were produced and analyzed.

### 2.8. Analysis of Mitochondrial Morphology

Mitochondrial morphology was quantitatively analyzed using a computer-assisted morphometric application to calculate the aspect ratio values, as described previously [23]. The acquired mitochondrial images were evaluated using ImageJ software (version 1.53). The aspect ratio values were derived from the lengths of the major and minor axes, and the average aspect ratio values were statistically calculated. A value of one indicated a perfect circle. As the mitochondria elongated and became more elliptical, the aspect ratio values increased.

### 2.9. Statistical Analysis

Data are presented as the mean ± standard error of the mean (SE) (n > 3). Differences among the treated groups were evaluated using one-way analysis of variance (ANOVA) with Tukey’s post hoc test using GraphPad Prism 9.12 software (GraphPad, San Diego, CA, USA).

## 3. Results

### 3.1. Expression Patterns of PA200 and Ecpas in Testes and Epididymides

PA200 and ECPAS bind the 20S CP and 26S proteasome, which assembled 19S with 20S, respectively (Figure 1a). PA200 and ECPAS are expressed in the mouse testes; however, their expression patterns along the transportation route from the testis to the epididymis were previously unknown. To this end, we performed immunoblotting on the testis and epididymis samples of mice. PA200 and ECPAS were both expressed in similar proportions in the testes and epididymides, except in the cauda (Figure 1b). To investigate the involvement of PA200 and ECPAS in spermatogenesis, we examined their expression in murine testes in more detail. The immunoblots of mouse whole-testis lysates from 1-, 2-, 3-, 4-, and 5-week-old mice corresponded to the stages of spermatogonia, zygotene spermatocytes, pachytene spermatocytes, round spermatids, and elongating spermatids, respectively [24]. Both proteins were detected in the testes of at least 1-week-old mice (Figure 1b).

To investigate their localization in testicular tissues during spermatogenesis and in mature sperm, sections of testes and epididymides were immunostained with anti-PA200 and anti-ECPAS Abs (Figure 1c). Both Abs reacted with testicular tissues. Throughout spermatogenesis, PA200 proteins were predominantly localized in the cytoplasm until the round spermatid stage, and to the head and midpiece in elongating and elongated spermatids (a set of upper images in Figure 1d). As previously reported [12], PA200 is involved in histone acetylation in the sperm nucleus during spermiogenesis. PA200 proteins are also expressed in the sperm midpiece, implying an unreported function.

Similar to PA200, ECPAS was detected in the cytoplasm of sperm and localized in residual bodies and the acrosome at the elongating spermatid stage. In the epididymides, PA200 and ECPAS were localized in the midpiece and acrosome, respectively (Figure 1d,e). Thus, these localizations were altered stepwise during sperm maturation, suggesting that they may work in a sequential manner (Figure 1e). Based on the above results, we hypothesized that PA200 and ECPAS are involved in spermiogenesis.

### 3.2. Fertility of dKO Male Mice

As previously reported, *Psme4* KO male mice were subfertile [12,18]. However, in this study, *Ecpas* KO male mice were fertile [22]. We speculated that PSME4 and ECPAS could cooperate functionally to maintain spermatogenesis. To explore this possibility, we generated double KO (dKO) mice. Heterozygous mice did not exhibit obvious abnormalities and were intercrossed to obtain dKO mice (Figure 2a,b). Although dKO females sustained normal fertility, dKO males were completely infertile despite normal mating behavior, as judged by the formation of copulation plugs (Figure 2c). To explore the cause of infertility in male dKO mice, the PAS staining of the male reproductive organs was performed. Although no notable morphological changes were observed in the testes, the number of sperm appeared to slightly decrease in the epididymides of the dKO mice (Figure 2d). These results support the possibility that PA200 and ECPAS cooperate functionally to maintain spermatogenesis.

### 3.3. LPIN1 as a Target Protein of PA200 and ECPAS

To evaluate proteasome activity, we determined the chymotrypsin-like activity in the testes and epididymides (Figure 3a). The activity in proteasomes was significantly reduced in dKO mice compared with wild-type (WT) mice (Figure 3b). To examine whether the detected activity was proteasome-dependent, MG132 (proteasome inhibitor) was added to the cell lysates. Since proteasome activity was decreased in dKO mice, we considered that specific target proteins, which were degraded by proteasomes with both PA200 and ECPAS proteins in WT mice, accumulated in the testes and epididymides of dKO mice.

To identify the predicted target proteins of proteasomes with PA200 and ECPAS, we performed iTRAQ-based proteomic analysis of the epididymal tissues of WT, *Psme4* KO, *Ecpas* KO, and dKO mice. We searched for proteins that significantly increased only in dKO mice, according to the following criteria: The difference between expression levels in dKO and WT tissues was an abundant ratio > 2, while those in control mice (*Psme4* KO and *Ecpas* KO) had an abundance ratio >0.8, <1.2 (Figure 3c). Fifteen proteins were identified (Appendix A). Among them, phosphatidate phosphatase LPIN1 exhibited the highest increase in dKO mice and accumulated in the testes (Figure 3d and Appendix A). Therefore, LPIN1 proteins were further analyzed.

LPIN1 catalyzes phosphatidic acid (PA) in diacylglycerol. *lpin1* deficiency in fatty liver dystrophy (*fld*) mutant mice causes male infertility and impairment of adipose tissue development [25]. The overexpression of lpin1 leads to the accumulation of triacylglycerol (TAG), which is the final product of PA [26,27,28]. To investigate lipid accumulation in dKO mice, frozen sections were obtained from the testes of WT and dKO mice and stained with Oil Red O to visualize lipid droplets (LDs) containing TAG in the cytoplasm. As expected, the deposition of Oil Red O in dKO testes was greater than that in WT testes (Figure 3e). LPIN1 phosphatase regulates mitochondrial dynamics, suggesting that both PA consumption and diacylglycerol production via LPIN1 promote mitochondrial fission [29]. To examine whether mitochondrial morphology or number was altered in the testes of dKO mice, we evaluated the mitochondria in spermatocytes using TEM. The number of mitochondria in dKO spermatocytes was significantly increased by 1.7-fold compared with that in WT spermatocytes (left graph in Figure 3f). To determine the dynamics of mitochondria in spermatocytes, we compared the aspect ratio of mitochondria between the WT and dKO testes. The aspect ratio of the mitochondria was comparable between them (1.1-fold), although there were significant differences in mitochondrial fission (right graph in Figure 3f). These results suggest that the accumulation of LPIN1 alters the number of mitochondria by promoting mitochondrial fission.

### 3.4. Loss of PA200 and ECPAS Causes Midpiece Defects in Sperm

During spermiogenesis, round spermatids undergo morphological alterations, forming mature sperm cells. These alterations include nuclear condensation, the formation of the mitochondrial sheath at the midpiece, and the elimination of the residual body to discard non-essential cytosolic material. During spermiogenesis, mature sperm are released into the seminiferous lumen. As shown in Figure 2c and Figure 3f, dKO mice were infertile, and their spermatocytes contained an increased number of mitochondria. This result reinforces the possibility that mature dKO spermatozoa show substantial abnormalities. To identify morphological changes in sperm from the cauda of dKO mice, we determined the length of the midpiece and tail of the sperm. As shown in Figure 4a, the flagellum length without a midpiece was comparable between WT and dKO mice. In contrast, the length of the midpiece was significantly shortened in dKO sperm (Figure 4b). To determine the abnormal appearance of the sperm from dKO mice, we evaluated the morphological appearance of the sperm collected from the dKO epididymis cauda, and multiple abnormalities were observed, specifically in the midpiece and head. Midpiece defects were significantly increased in the sperm (65.3%) of dKO mice. Concurrently, a double head was present (12.3%), suggesting a lack of proper cell separation (Figure 4c,d). In addition, the malformed sperm in dKO mice lacked motility (Appendix A). TEM analysis confirmed abnormal multiple flagella within the same membrane and abnormal positioning of the mitochondria (Figure 4e). Finally, to investigate alteration in mitochondrial morphology, we examined the aspect ratio of mitochondria in longitudinal and transverse sections of sperm. The mitochondrial sheath in the midpiece of sperm has a normally uniform ellipse in longitudinal sections because it tightly coils around the midpiece of the sperm flagellum [30,31]. The aspect ratio of mitochondria in longitudinal sections was significantly decreased in dKO sperm compared with that in the WT sperm (Figure 4f). Furthermore, the aspect ratio of mitochondria in transverse sections was significantly decreased in sperm from dKO mice (Figure 4g). Therefore, a reduction in the aspect ratio in both sections was expected to reduce the size of the mitochondria in dKO sperm. These results suggest that morphological alteration in mitochondria promotes mitochondrial fission caused by a defective midpiece in the sperm from dKO mice.

## 4. Discussion

Since the two homologous proteasome regulators, PA200 and ECPAS, share a HEAT-like repeat motif, and deficient mice exhibit no noticeable abnormality in fertility, we raised the possibility that these two proteins functionally compensate for reproductive ability. In this study, we revealed their expression patterns in sperm along their transportation route from the testes to the epididymides (Figure 1d). Further studies using post-meiotic markers, such as AKAP4 [32,33,34], are necessary to clarify the detailed expression patterns of PA200 and ECPAS in the testis depending on cell type. In this study, we demonstrated that (1) dKO mice exhibited complete male infertility due to malformation in the midpiece of the sperm, and (2) the accumulation of LPIN1 in dKO sperm caused the deposition of LDs and enhanced the number of mitochondria. Subsequently, our results indicate that PA200 and ECPAS play a key role in spermatogenesis and may be critical for appropriate amounts of LPIN1 proteins during morphological changes in mitochondria associated with the completion of spermiogenesis and the formation of mature midpieces (Figure 5).

Proteasomes consist of several components, including CP, RP, and regulatory proteins. Furthermore, CP constitutes a specialized form of the spermatoproteasome, which contains a4s in the male reproductive organs. Recently, several studies have demonstrated that various regulatory factors and a4s in spermatoproteasomes play critical roles in spermatogenesis [8,9,20]. Male mice lacking *Psma8*, a gene encoding a4s, are infertile because of the absence of a proper exit from meiosis I during spermatogenesis [35]. The polyubiquitin gene *Ubb* is also required for exit from meiosis I in both males and females [36]. Recently, the deletion of *Ubb* was reported to be closely related to PSMA8 protein expression [37]. In addition, we have previously reported that dKO male mice lacking *Psme3* and *Psme4* genes are also completely infertile because of sperm suffering from oxidative stress damage [19]. However, although most of these deficient murine models produce problematic proteasomes, their male fertility and sperm morphology are normal.

Here, we revealed that dKO sperm from mice harboring *Psme4* and *Ecpas* genes displayed malformed sperm, such as defective midpieces and double heads, due to abnormalities in late spermatogenesis (spermiogenesis) (Figure 4). Therefore, based on our findings and previous studies, we hypothesized that PSME4- and ECPAS-driven pathways could be functional at a later stage during spermatogenesis. Although malformed sperm, as mentioned above, are often observed in infertile male patients, the onset mechanism of malformed sperm is poorly understood. In contrast, male mice lacking DCUN1D1, an E3 component of the neddylation factor, also yield midpiece defects and double heads [38]. Furthermore, mice lacking PITHDI, a proteasome-interacting protein, exhibit a corresponding sperm abnormality [20]. Based on these and our present findings, we assumed that the functions of PSME4 and ECPAS might overlap as target proteins during spermiogenesis. The ultrastructural morphological abnormality in the mitochondrial sheath is one of the causes of reduced sperm motility and matches the clinical features of patients with teratozoospermia. In this study, we propose that proteasomes with PA200 and ECPAS play a role in mitochondrial sheath formation in sperm by regulating LPIN1 quantity. Mice lacking Vps13a (Vaculor protein 13 homolog a) and seipin exhibit sperm abnormalities similar to our results, including morphological differences in TEM and light microscopy and reproductive ability [39,40]. Although these genes are associated with lipid metabolism in mice, the detailed mechanisms of lipid metabolism and mitochondrial sheath formation are still unknown.

Mitochondrial fission and fusion are highly regulated by specific proteins and are important for controlling the mitochondrial shape, distribution, size, and number. The fusion factor (Mfn; Mitofusin) and fission factor (Mff; mitochondrial fission factor) are required for spermatogonial differentiation and organization of the mitochondrial sheath in sperm, respectively [41,42]. Since these proteins are recruited by specific membrane lipids on mitochondria, the membrane lipid composition in mitochondria is involved in mitochondrial dynamics. Phosphatidic acid (PA), a substrate for LPIN1, promotes mitochondrial fusion by inducing membrane curvature in conjugation with MFN tethering. Diacylglycerol (DAG), which catalyzes PA via LPIN1, promotes mitochondrial fission by recruiting fission-promoting proteins. In this study, we showed that the accumulation of LPIN1 causes a disorganized mitochondrial sheath by enhancing mitochondrial fission. Thus, in support of our findings, impairment of mitochondrial dynamics causes disorganization of the mitochondrial sheath in the sperm. Since LPIN1 regulated lipid metabolism through proteasome control, the dKO mice in this study may provide novel insights into spermatogenesis in the future. In addition, we revealed that dKO sperm were immotile (Appendix A). We speculate that sperm motility may be reduced by disorganizing the mitochondrial sheath structure. Mice lacking glycerol kinase family member proteins exhibit low sperm motility and sperm abnormalities of the mitochondrial sheath [30,43]. Therefore, our findings and previous reports suggest that the rigorous formation of the mitochondrial sheath is essential for sperm motility. Here, we propose that the PA200- and ECPAS-regulated fine-tuning of specific proteins by proteasomes is required to maintain the proper number of mitochondria in the sperm. Further analysis is required to confirm this hypothesis.

Multiple morphological abnormalities of the sperm flagella (MMAF) phenotype, a severe form of asthenoteratozoospermia, are now known as a major reason for male infertility [44]. Our results highlighted an essential role for PA200 and ECPAS in spermatogenesis and male fertility and showed that dKO leads to the MMAF phenotype, thus improving our understanding of the mechanisms underlying sperm mitochondrial sheath formation during spermiogenesis.

In conclusion, our results suggest that proteasomes with PA200 and ECPAS play a role in the formation of the mitochondrial sheath in sperm by regulating LPIN1 quantity. The reduction in sperm motility observed in male mice with dKO indicates that alterations in proteasome activity may affect the mitochondrial sheath’s structure. Future investigations on dKO mice are needed to understand the underlying mechanisms of PA200 and ECPAS in spermatogenesis.

## Figures and Tables

**Figure 1 biomolecules-13-00586-f001:**
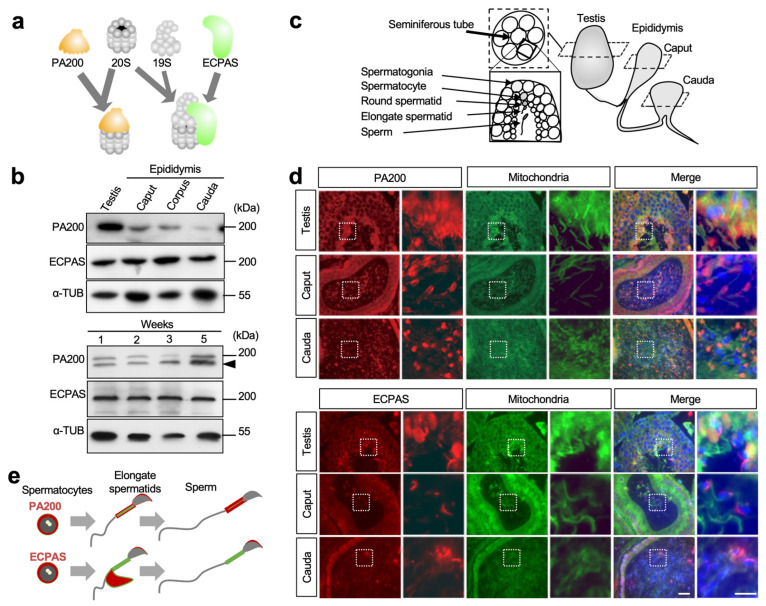
PA200 and ECPAS expression patterns in male reproductive organs: (**a**) Schematic representation of proteasome regulators. PA200 (orange) directly binds to 20S core particle (CP) and controls protease activity. ECPAS (green) binds to 26S proteasome and regulates its assembly; (**b**) Expression pattern of PA200 and ECPAS in the testes and epididymides. Arrowhead: non-specific PA200 band; (**c**) Schematic representation of male reproductive organs. For immunostaining, sections from each organ were used (broken lines). Spermatogenesis progresses from the outside to the inside of the seminiferous tubes; (**d**) Fluorescent images of the testes and epididymides stained with anti-PA200 Ab (red), anti-ECPAS Ab (red), MitoTracker (green), and DAPI (blue): (left) scale: 20 µm; (right) enlarged images of dashed boxes at left; scale: 10 µm; (**e**) Image showing the localization of PA200 and ECPAS during spermatogenesis.

**Figure 2 biomolecules-13-00586-f002:**
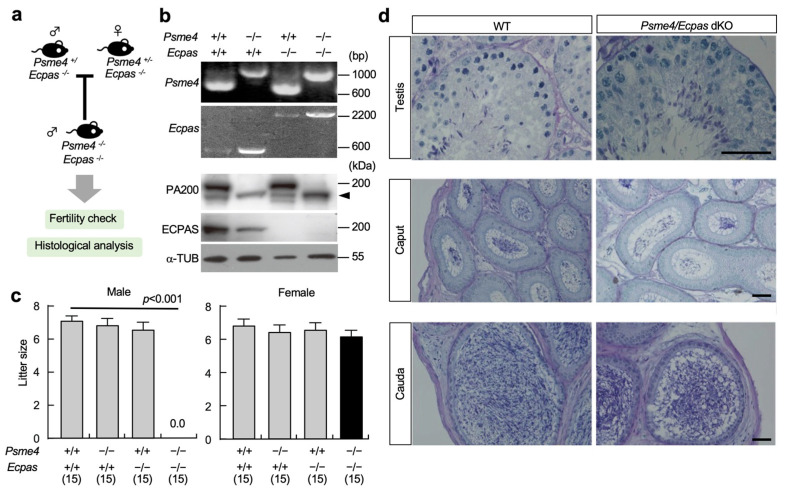
Generation of *Psme4/Ecpas* double knockout (dKO) and male infertility: (**a**) Experimental flow of generation and analysis of dKO mouse; (**b**) Genotyping and immunoblotting of *Psme4* (PA200) and *Ecpas* (ECPAS) in dKO mice. Arrowhead: non-specific band of PA200; (**c**) Fecundity of male and female dKO mice (*n* = 15). Data are expressed as the mean ± standard error; (**d**) PAS staining images of testes and epididymides in wild-type (WT) and dKO mice. Scale: 40 µm.

**Figure 3 biomolecules-13-00586-f003:**
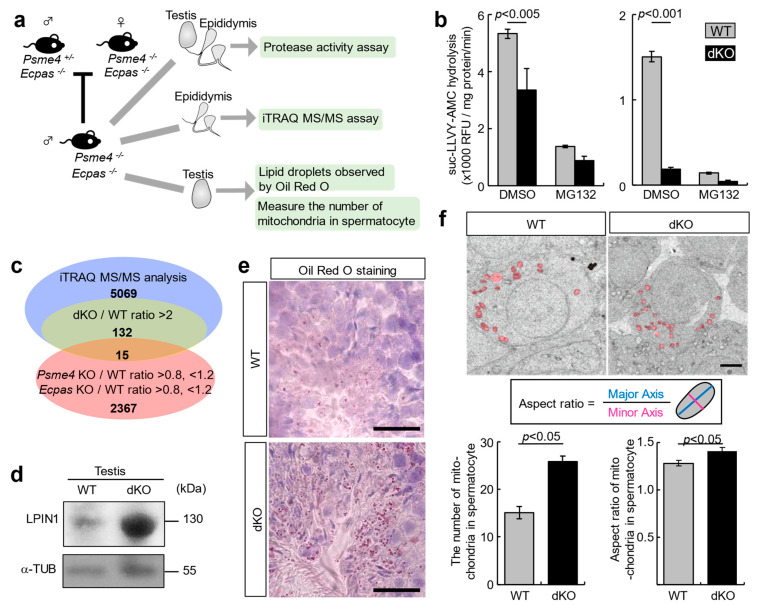
Impaired proteasome activity accumulates LPIN1 in the testes of *Psme4/Ecpas* double-knockout (dKO) mice: (**a**) Experimental flow of identification of target protein of proteasome involved with PA200 and ECPAS; (**b**) The chymotrypsin-like activities of the proteasome in the *Psme4/Ecpas* dKO testes (left) and epididymides (right) (*n* = 3). MG132 effectively blocks the proteolytic activity of the 26S proteasome complex. Data are expressed as the mean ± standard error; (**c**) Detection of accumulated proteins in the *Psme4/Ecpas* dKO epididymides by iTRAQ MS/MS analysis; (**d**) Immunoblotting of LPIN1 in dKO testes; (**e**) Testicular sections stained with hematoxylin and eosin (HE) and Oil Red O; scale: 20 μm; (**f**) The appearance of mitochondria in dKO spermatocyte: (upper) transmission electron microscopy (TEM) images of mitochondria (red) in dKO spermatocyte; scale: 2 μm; (lower) the number of mitochondria (*n* = 30) and the aspect ratio of mitochondria (*n* = 45) in spermatocytes. Data are expressed as the mean ± standard error.

**Figure 4 biomolecules-13-00586-f004:**
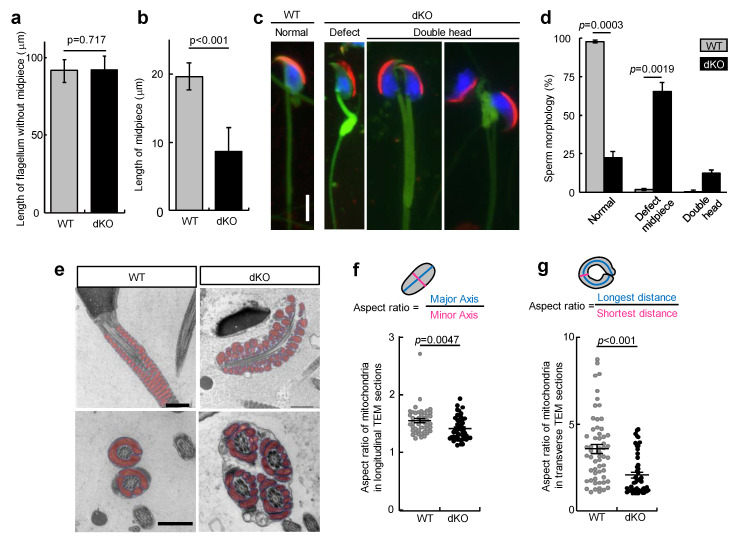
Mitochondrial defects of epididymal sperm of *Psme4/Ecpas* dKO mice: (**a**) Total flagella length without midpiece of sperm. Data are expressed as the mean ± standard error. (**b**) The midpiece length of sperm. Data are expressed as the mean ± standard error; (**c**) Fluorescent images of sperm’s head and midpiece stained with PNA (red), DAPI (blue), and MitoTracker (green); scale: 5 μm; (**d**) The ratio of sperm with defective midpiece and double head. Data are expressed as the mean ± standard error; (**e**) Transmission electron microscope (TEM) images of the sperm midpiece mitochondrion (highlighted in red): (upper) longitudinal section; (lower) transverse section; scale: 1 μm; (**f**) Aspect ratio of sperm mitochondria in longitudinal TEM sections. Scatter plot is individual values (*n* = 45). Lines and error bars expressed as the mean  ±  standard error; (**g**) Aspect ratio of sperm mitochondria in transverse TEM sections. Scatter plot is individual values (*n* = 57). Lines and error bars are expressed as the mean  ±  standard error.

**Figure 5 biomolecules-13-00586-f005:**
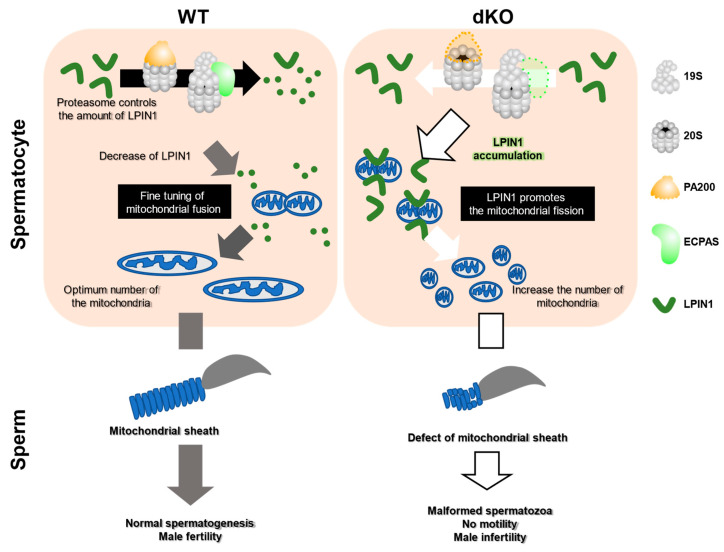
Schematic model of LPIN1 accumulation during spermatogenesis in dKO mouse. Among spermatogenesis, the proteasome controls the amount of LPIN1. At the mitochondrial fusion step, the proteasome may fine-tune the number of mitochondria via LPIN1 degradation which makes the number of mitochondria optimum. In contrast, dKO of PA200 and ECPAS (dashed lines) cause accumulation of LPIN1 in spermatocytes. Accumulated LPIN1 then promotes mitochondrial fission via the consumption of phosphatidic acid and increases the number of mitochondria in spermatocytes. Sperm in dKO showed a defective mitochondrial sheath and reduced motility.

## Data Availability

The data supporting the findings of this study are available from the corresponding author upon request.

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
