# Peer review of "Proteasome-Associated Proteins, PA200 and ECPAS, Are Essential for Murine Spermatogenesis"

_biomolecules, 2023, doi:10.3390/biom13040586_

Round 1

Reviewer 1 Report

Review: Proteasome-associated proteins, PA200 and ECPAS, are essen-2 tial for murine spermatogenesis

The present work by Sato and collaborators investigate the hypothesis that proteasome-associated proteins, PA200 and ECPAS, can complement each other during spermatogenesis, and they produce mice lacking both of these genes (double-knockout mice;).

The authors used complementary and extensive methods to evaluate the lack of these two genes that result in terms of infertility. Using proteomic, ultrastructural and microscopic analyses they demonstrated that PA200 and ECPAS 28 may work cooperatively during spermatogenesis and then are essential for male fertility.

Moderate english changes are required more specifically in the abstract.

The manuscript could be accepted for publication but only after minor revision and text corrections from the authors.

Major concerns:

As said by the authors, the expression of PA200 and ECPAS were already know to be expressed in the mouse testes, the authors showed for the first time their expression patterns along the transportation route from the testis to the epididymis.

However, since the authors used immunolabelling to identify sperm cell categories, we suggest that they should characterize them more clearly by using a specific post-meiotic marker such as AKAP4 protein (Pelz et al. 2017 ; Nixon et al. 2019 ; Ernst et al. 2019 ; Scovell et al. 2021) to be more rigorous on spermiogenesis expression by immunolabeling (or western blotting). The expression of AKAP4 (precursor and processed forms) begins in the cytoplasm of round spermatids, then in elongated spermatids, and in the flagellum of spermatozoa before to be released in the seminiferous tubes, and is then a good candidate to be used in spermatogenesis description and identify mature sperm in such a study.

Furthermore, in their morphological description, the authors only described the midpiece region and as said, it will be interested to have labeling of the principal of the flagellum (with the fibrous sheath) that is the other structure, to have a complete overview of the expression and morphology of the sperm cells in the double Ko mice.

The manuscript could be then accepted for publication after this minor revision and text corrections from the authors.

Author Response

Response to Reviewer 1 Comments

Comment 1: The authors used complementary and extensive methods to evaluate the lack of these two genes that result in terms of infertility. Using proteomic, ultrastructural and microscopic analyses they demonstrated that PA200 and ECPAS 28 may work cooperatively during spermatogenesis and then are essential for male fertility

Response: First of all, we appreciate your thoughtful comment.

Comment 2: Moderate english changes are required more specifically in the abstract.

Response: Our manuscript has been proofread by native speakers. We attached a certification of proofreading company in the end of this document.

Comment 3: As said by the authors, the expression of PA200 and ECPAS were already know to be expressed in the mouse testes, the authors showed for the first time their expression patterns along the transportation route from the testis to the epididymis. However, since the authors used immunolabelling to identify sperm cell categories, we suggest that they should characterize them more clearly by using a specific post-meiotic marker such as AKAP4 protein (Pelz et al. 2017 ; Nixon et al. 2019 ; Ernst et al. 2019 ; Scovell et al. 2021) to be more rigorous on spermiogenesis expression by immunolabeling (or western blotting). The expression of AKAP4 (precursor and processed forms) begins in the cytoplasm of round spermatids, then in elongated spermatids, and in the flagellum of spermatozoa before to be released in the seminiferous tubes and is then a good candidate to be used in spermatogenesis description and identify mature sperm in such a study.

Response: Thank you for your useful comment. As the reviewer suggested, our experiment could not identify the expression period of PA200 and ECPAS in testes. The manuscript may be improved by additional experiments which use a specific post-meiotic marker such as AKAP4 protein. However, our findings about PA200 and ECPAS expression in testes were not the main argument in this study. Our novel finding was the localization of PA200 and ECPAS in matured sperm inside epididymis. Since the reviewer’s point is a good suggestion, we add new sentences to the discussion section as below: In this study, we revealed their expressions patterns in sperm along their transportation route from the testes to the epididymis (Fig. 1d). Detailed expression patterns of these proteins in testis depending on cell type were not proved in this study. In the future, further experiments should use post-meiotic markers, such as AKAP4 [33-35]. (page8 line295)

Comment 4: Furthermore, in their morphological description, the authors only described the midpiece region and as said, it will be interested to have labeling of the principal of the flagellum (with the fibrous sheath) that is the other structure, to have a complete overview of the expression and morphology of the sperm cells in the double Ko mice.

Response: Thank you for your valuable comment. As the reviewer mentioned, we incorrectly used the word “sperm tail” instead of “the flagellum length without midpiece”. As shown in Fig4a, the length of the flagellum without midpecice is not different between WT and KO mice.  Accordingly, we have also changed the words, “tail” to “flagellum length without a midpiece” (page 7, line 257, Y-axis in Fig. 4a).

Reviewer 2 Report

In this paper, Sato et al. proposed that PA200 and ECPAS 28 work cooperatively during spermatogenesis and are essential for male fertility.

The paper is interesting.

However, Authors should argue in a better way the relevance of their results and their implications in the diagnosis or treatment of male infertility.

Therefore, I suggest to better organize the discussion section (paragraphs 4.1-4.3) in a more organic manner (without paragraphs). 

Authors should discuss their main results:

- “proteasomes with PA200 and ECPAS play a role in the formation of the mitochondrial sheath in sperm by regulating LPIN1 quantity”;

- “the reduction in sperm motility observed in male mice with dKO indicates that alterations in proteasome activity may affect the mitochondrial sheath’s structure”;

by comparing them with the work that was done by others and pointing out the things that their study does which was never done before

Finally, Authors should connect molecular defects with their implications in the diagnosis or treatment of male infertility.

Author Response

Response to Reviewer 2 Comments

Comment 1: In this paper, Sato et al. proposed that PA200 and ECPAS 28 work cooperatively during spermatogenesis and are essential for male fertility.

The paper is interesting.

However, Authors should argue in a better way the relevance of their results and their implications in the diagnosis or treatment of male infertility.

Response: First of all, we appreciate your thoughtful comment.

Comment 2: Therefore, I suggest to better organize the discussion section (paragraphs 4.1-4.3) in a more organic manner (without paragraphs). 

Authors should discuss their main results:

- “proteasomes with PA200 and ECPAS play a role in the formation of the mitochondrial sheath in sperm by regulating LPIN1 quantity”;

- “the reduction in sperm motility observed in male mice with dKO indicates that alterations in proteasome activity may affect the mitochondrial sheath’s structure”;

by comparing them with the work that was done by others and pointing out the things that their study does which was never done before

Finally, Authors should connect molecular defects with their implications in the diagnosis or treatment of male infertility.

Response: I agree with your comment. Thank you for the useful comment. According to the reviewer’s comment, we have re-organized the discussion section and removed the subheading from the discussion section. We have added new sentences (page 10 line 359, line 361, and 370).